# Staple Food Preference and Obesity Phenotypes: The Regional Ethnic Cohort Study in Northwest China

**DOI:** 10.3390/nu14245243

**Published:** 2022-12-09

**Authors:** Kun Xu, Binyan Zhang, Yezhou Liu, Baibing Mi, Yutong Wang, Yuefan Shen, Guoshuai Shi, Shaonong Dang, Xin Liu, Hong Yan

**Affiliations:** 1Key Laboratory for Disease Prevention and Control and Health Promotion of Shaanxi Province, Department of Epidemiology and Biostatistics, Global Health Institute, School of Public Health, Xi’an Jiaotong University Health Science Center, 76 West Yanta Road, Xi’an 710061, China; 2Nutrition and Food Safety Engineering Research Center of Shaanxi Province, Key Laboratory of Environment and Genes Related to Diseases, Xi’an Jiaotong University, 76 West Yanta Road, Xi’an 710061, China

**Keywords:** obesity phenotypes, food preference, rice, wheat

## Abstract

Staple food preference vary in populations, but evidence of its associations with obesity phenotypes are limited. Using baseline data (*n* = 105,840) of the Regional Ethnic Cohort Study in Northwest China, staple food preference was defined according to the intake frequency of rice and wheat. Overall and specifically abdominal fat accumulation were determined by excessive body fat percentage and waist circumference. Logistic regression and equal frequency substitution methods were used to evaluate the associations. We observed rice preference (consuming rice more frequently than wheat; 7.84% for men and 8.28% for women) was associated with a lower risk of excessive body fat (OR, 0.743; 95%CI, 0.669–0.826) and central obesity (OR, 0.886; 95%CI, 0.807–0.971) in men; and with lower risk of central obesity (OR, 0.898; 95%CI, 0.836–0.964) in women, compared with their wheat preference counterparties. Furthermore, similar but stronger inverse associations were observed in participants with normal body mass index. Wheat-to-rice (5 times/week) reallocations were associated with a 36.5% lower risk of normal-weight obesity in men and a 20.5% lower risk of normal-weight central obesity in women. Our data suggest that, compared with wheat, rice preference could be associated with lower odds ratios of certain obesity phenotypes in the Northwest Chinese population.

## 1. Introduction

Obesity, characterized by abnormal or excessive fat accumulation, is the leading risk factor for type 2 diabetes, cardiovascular disease, and a number of chronic illnesses [1]. Central obesity significantly increases the risk of incident diabetes [2], specific cancers [3], and all-cause mortality [4] independent of body mass index (BMI), which is used to define overweight and general obesity widely. In addition, recently observed obesity phenotypes such as normal-weight obesity (NWO) and normal-weight central obesity (NWCO) underlined the risks of diabetes, cardiovascular mortality, and all-cause mortality for people commonly classified as normal weight [5,6,7,8]. Moreover, studies indicated the varied metabolic status of different obesity phenotypes [9,10,11], yet the determinants of specific adiposity phenotypes remain unclear.

Abnormal or excessive fat accumulation may result from a set of intricate interactions between the environment, genetic predisposition, and human behaviors [12]. Dietary habits, such as high caloric diet, sugar-sweetened beverages intake, and western dietary patterns, are reported risk factors [13,14]. Recently, an increasing number of studies suggested that food preferences and choices were also closely associated with obesity [15,16]. In China, more than half of adults had overweight or general obesity, and 40.9% had central obesity [14,17]. Rice and wheat contribute a great majority of calories in Chinese diets [18,19], while less is known about the associations between staple food preference and obesity phenotypes in Chinese.

The demand for wheat consumption is increasing globally, even in regions where the climate is not suitable for wheat production [20]. Traditionally, wheat products, such as noodles and steamed bread, are the main staple food in north China, while steamed rice is the main staple food in south China [21]. With the increase in commodity circulation and personnel flow, both rice and wheat have been available and acceptable for Chinese residents [19]. Previous studies have suggested that rice intake was negatively associated with obesity, while high intake of wheat showed the opposite associations [22,23,24], but the potential effects of the staple food preference on obesity phenotypes were not well understood. In the present study, we aimed to evaluate the associations of staple food preference with several obesity phenotypes in a large sample of northwest China.

## 2. Materials and Methods

### 2.1. Study Design and Population

Our study used the baseline data from the Regional Ethnic Cohort Study in Northwest China (RECS), an ongoing prospective cohort study established among residents of five northwest provinces (Shaanxi, Gansu, Qinghai, Ningxia, and Xinjiang). Details of the study design have been described elsewhere [25]. In brief, 118,572 participants aged 35–74 were recruited between June 2018 and May 2019. Information on demographic characteristics (such as sex, date of birth, ethnic group, educational level, and family income), lifestyle factors (such as alcohol consumption, smoking, dietary status, and physical activity), and medical history were collected by questionnaires via face-to-face interviews by well-trained technicians at the local clinical center. Anthropometric measurements were conducted by qualified nurses or healthcare workers, and all devices were calibrated. Standing height was measured by a stadiometer when participants were wearing light clothes and without shoes. Subsequently, body weight and body fat percentage (BFP) were measured by bioelectrical impedance analysis instrument (Tanita DC-430MA). Waist circumference (WC) was measured to the nearest 0.1 cm at the midpoint between the lowest rib and the iliac crest using a measuring tape. BMI was calculated as body weight in kilograms divided by squared height in meters. In order to assess the associations of staple food preference with obesity phenotypes, participants with diabetes (*n* = 5739), cancers (*n* = 616), and those with missing values (*n* = 3633, for missing intake frequency of rice or wheat; *n* = 2744, for missing all three indices [BFP, BMI, and WC]) were excluded, leaving 105,840 (men, 42,596 and women, 63,244) participants in the analyses.

This study was approved by the Ethical Committee of Xi’an Jiaotong University Health Science Center, and all the participants provided written informed consent.

### 2.2. Assessment of Staple Food Preference

We used an adapted version of the food frequency questionnaire in the China Kadoorie Biobank, which was shown to have good validity and reproducibility [26]. Participants were required to select one of five options (daily, 4 to 6 times per week, 1 to 3 times per week, 1 to 3 times per month, or never or rarely) based on their dietary habits during the past 12 months. The staple food preference category was developed according to the weekly intake frequency of rice and wheat: rice as a staple food (consuming rice every day or 4–6 times per week, and consuming wheat less than 4–6 times per week); wheat as a staple food (consuming wheat every day or 4–6 times per week, and consuming rice less than 4–6 times per week); both rice and wheat as a staple food (the rest of the participants), which means that they consumed rice and wheat with similar frequency.

### 2.3. Definition of Outcomes

Excessive body fat was defined as BFP > 25% for men or >35% for women [27]. Central obesity was defined as WC ≥ 90 cm for men or ≥80 cm for women [28]. Furthermore, NWO or NWCO was defined as BMI within the normal range (18.5–24.9 kg/m^2^) but with excessive body fat or central obesity.

### 2.4. Statistical Analysis

Mean and standard deviation (SD) for continuous variables or number and percentage for categorical variables were used to describe population characteristics by staple food preference and sex. Analysis of variance (ANOVA) was used to compare differences in population characteristics across the staple food preference for continuous variables, and the chi-square test was used for categorical variables. Odds ratios (ORs) and its 95% confidence intervals (95%CIs) were estimated by logistic regression for obesity phenotypes according to intake frequency of rice or wheat (never or rarely and 1–3 times/month were combined, considering the low proportion of participants who never or rarely consumed wheat, Appendix A), or staple food preference categories by sex. Model 1 was fitted with the adjustment of age, ethnicity (Han, Hui, Uygur, and Others), and province. In model 2, adjustment was made further concerning alcohol drinking (every week or every month and occasionally or never), smoking status (every day or most days and occasionally or never), family income (less than 20,000, 20,000–50,000, higher than 50,000 yuan/year, or do not know), and education levels (primary school or less, middle or high school, or college or university); and in model 3, additional adjustment involving physical activity (metabolic equivalent of task hours (MET-hr)/day) was made. In stratification analyses, categorical variables for stratifying were not adjusted in the models correspondingly. Percentages of missing values in covariates were less than 1%, except for family income (1.4%) and physical activity (2.6%). Random forest algorithm (R package, “mice”) was used for the imputation of all missing values. In addition, the equal frequency substitution method was used to evaluate the associations of staple food preference with obesity phenotypes by reallocating two staple foods (5 times/week) from wheat to rice by analogy with the principle and method of isotemporal substitution analysis [29].

In this study, we used R Software (version 4.1.0; https://www.r-project.org (accessed on 14 July 2021)) for all the analyses and figures plotting, and a two-tailed *p* < 0.05 was considered statistically significant.

## 3. Results

In our study population, 7.84% of men and 8.28% of women preferred rice as a staple food, 48.71% of men and 49.91% of women had excessive body fat, while 41.20% of men and 63.74% of women had central obesity. Participants preferring rice as staple food tended to be younger, have higher education levels, lower physical activity, higher family income, and were more likely to be Han or regular alcohol drinkers, similarly for men and women (Table 1).

The associations of wheat intake frequency, rice intake frequency, and staple food preference with excessive body fat and central obesity were displayed in Figure 1. Higher wheat intake was associated with higher risks of excessive body fat and central obesity in men and central obesity in women. ORs (95%CIs) for comparing extreme wheat intake frequency were 1.575 (1.365, 1.819) for excessive body fat, 1.241 (1.087, 1.418) for central obesity in men, and 1.129 (1.028, 1.239) for central obesity in women, with the adjustment of age, ethnic, province, drinking status, smoking status, family income, education level, and physical activity (MET-hr/day). However, the associations of rice intake frequency with excessive body fat and central obesity were not obvious, which tended to be positive in men and negative in women. Inverse associations of staple food preference trends from wheat and both to rice were observed for excessive body fat in men (*p* < 0.001) and central obesity in women (*p* = 0.014). ORs (95%CIs) for rice preference were 0.743 (0.669, 0.826) for excessive body fat, 0.886 (0.807, 0.971) for central obesity in men, and 0.898 (0.836, 0.964) for central obesity in women, compared with their wheat preference counterparties respectively.

In participants with normal BMI (Figure 2), these associations were similar but stronger. ORs (95%CIs) for comparing extreme wheat intake frequency were 2.215 (1.677, 2.926) for NWO in men and 1.262 (1.116, 1.425) for NWCO in women. Higher rice intake was significantly associated with lower risks of NWCO in women (OR and 95%CI for comparing extreme rice intake frequency was 0.728, 0.638–0.830). Additionally, rice preference was associated with a lower risk of NWO (OR, 0.571; 95%CI, 0.471–0.693) and NWCO (OR, 0.778; 95%CI, 0.647–0.937) in men and with a lower risk of NWO (OR, 0.851; 95%CI, 0.748–0.967), and NWCO (OR, 0.769; 95%CI, 0.700–0.845) in women, compared with their wheat preference counterparties.

In addition, replacing wheat intake (5 times/week) for equivalent frequency of rice was significantly associated with a lower risk of excessive body fat (OR, 0.806; 95%CI, 0.743–0.876) in men and central obesity (OR, 0.918; 95%CI, 0.865–0.975) in women. In participants with normal BMI, wheat-to-rice (5 times/week) reallocations were associated with a 36.5% lower risk of NWO in men, a 15.7% lower risk of NWO, and a 20.5% lower risk of NWCO in women (Table 2).

Consistent inverse associations were observed for the associations of rice preference with excessive body fat and NWO in men (Appendix A) and with central obesity and NWCO in women (Appendix A) in stratification analyses by age group, alcohol drinking status, smoking status, physical activity level, education level, and family income. Similar but stronger associations can be observed in participants with normal BMI.

Inverse associations or trends in staple food preferences (wheat, both wheat and rice, and rice) with excessive body fat in men were observed in Shaanxi and Gansu, while in Ningxia, a positive association was observed, and no significant association in Xinjiang (Appendix A). In contrast, the associations with central obesity in women were only significant in Shaanxi (Appendix A). With our participants, rice and wheat intake frequency varied across provinces (Appendix A). The proportion of participants consuming wheat every day ranged from 65.5% in Shaanxi to 98.3% in Ningxia, and that of consuming rice every day ranged from 6.7% in Xinjiang to 71.2% in Ningxia.

A sensitive analysis was conducted by defining excessive body fat as BMI > 25 kg/m^2^, and the results were similar but weaker than those of BFP (Appendix A).

## 4. Discussion

In the present study with 105,840 middle-aged and elderly Chinese, we observed inverse associations of rice preference with excessive body fat in men and with central obesity in women when compared with their wheat preference counterparties, respectively. In addition, similar but stronger inverse associations were observed in participants with normal BMI. Replacing wheat intake (5 times/week) with the equivalent frequency of rice was significantly associated with a 36.5% lower risk of NWO in men and a 20.5% lower risk of NWCO in women. To the best of our knowledge, this is the first study to report the associations of staple food preference with risks of certain obesity phenotypes.

Rice and wheat are the staple food indispensable in the Chinese diet. In line with our findings, previous studies in Chinese populations have shown that identified dietary patterns with high rice intake were inversely associated with obesity, while dietary patterns with the high wheat intake increased the risk of obesity [21,31]. In addition, a recent study in older adults (≥80 years) suggested that those who chose wheat as staple food had higher WC and BMI in men but not in women [32]. Our findings further indicated that, compared with wheat, rice preference or replacing wheat intake with the equivalent frequency of rice consumption was associated with a lower risk for several obesity phenotypes. Although the potential mechanism is still not clear, the underlying rationale may include the nutritional components, cooking methods, and dietary intake habits of rice and wheat. As the main staple food, rice and wheat serve as important sources of carbohydrates, proteins, fibers, necessary vitamins, and various minerals [33]. However, experiments based on animal models indicated that wheat gluten, a kind of protein, promoted weight gain through reducing thermogenesis and energy expenditure [34], while rice protein suggested the potential of anti-adiposity and triglyceride-lowering action by upregulating lipolysis and downregulating lipogenesis [35]. Moreover, wheat flour absorbs less water than rice when cooked, partly resulting in a higher energy density of wheat than rice. According to the Chinese Food Composition Table, 6th Edition 2020, the energy of noodles or steamed bread is twice or even three times that of steamed rice for the same weight [36]. It has been well demonstrated that dietary energy density is an independent predictor for obesity [37]. What is more, a study conducted in the US reported that rice consumers consumed more iron, potassium, dietary fiber, and vegetables but a smaller energy share of fat and saturated fat [38]. In China, generally, people with rice as a staple food are prone to eat several dishes with rice, always including vegetables, legumes, meat, or fish. However, people with wheat preference in northwest China tend to eat a large number of noodles with very little vegetables or meat in a meal, and sometimes with a large amount of oil, such as “You Po noodle” in Shaanxi [39]. Studies also indicated that rice-based dietary patterns often have high factor loadings of fresh vegetables, legumes, and fish [22,31], which were linked to lower adiposity risk in a systematic review [40]. However, the associations were different across provinces. Considering the vast territory and numerous ethnicities in northwest China, regional specificity and the potential underlying reasons for these associations may need further studies to explore.

Interestingly, the protective associations of rice vs. wheat preference were more remarkable for overall fat accumulation in men and abdominal fat accumulation in women. Gender-dependent variability of body fat distribution has been reported previously, and it could be explained by the interactions of environmental exposures with gene expression, sex hormones, and metabolic status. For example, studies have suggested that a common gene, CYP11*β*2, showed the gender difference in the interaction of sodium intake and obesity [41], and sex hormones have been indicated as playing an important role in the regional body fat distribution [42]. Furthermore, it has been reported that gender heterogeneity exists in the relationship between dietary patterns and obesity [22]. The physiological and pathological mechanisms of gender difference in the associations of habitual dietary intake with obesity phenotypes may need further studies to explore. Our results showed that rice preference was inversely associated with excessive body fat in men and with central obesity in women, which may offer insights into public health recommendations, considering that the prevalence of female central obesity is higher than that of male (women, 54.4%; men, 39.1%) and the higher increasing rate of excessive body fat in men vs. women, according to the China Health and Nutrition Survey [43].

BFP was used as the main indicator of excess body fat in this study, and great attention was paid to NWO and NWCO. As a proxy of body fat measurement, BMI cannot capture information about body composition and body fat distribution. Therefore, there is still a certain proportion of individuals gaining excessive body fat accumulation and/or central obesity even with normal BMI [5,44]. A cohort study of the US general people has found that individuals with NWCO had a greater total mortality risk than those with obesity defined by BMI [7], and the findings have been replicated by another cohort of Health Survey for England and the Scottish Health Survey [8]. Additionally, a large body of studies based on Chinese populations has also demonstrated that individuals with NWO or NWCO had a higher risk of hypertension, cardiometabolic dysregulation, and diabetes [5,45,46]. Our findings suggested that the inverse association of rice preference with NWO and NWCO was stronger than that with excessive body fat and central obesity, which highlighted the beneficial role of rice preference in individuals before their BMI increases as overweight or obesity.

Several limitations should be considered in the present study. Firstly, since the data on dietary intake were self-reported, recall bias could be an issue. Moreover, there was no information to identify the difference between refined and whole grain, which may be a confounding factor. Secondly, our study cannot ensure the causality of the observed associations because of the cross-sectional design. Finally, although we controlled for many known and available confounders, there would still be some potential factors not being included in our analyses, such as caloric intake. Despite these limitations, we have conducted sensitivity analyses, subgroup analyses, and equal frequency substitution analyses to ensure the robustness of the associations between staple food preference and obesity phenotypes.

## 5. Conclusions

Our study suggested that, compared with wheat preference, rice preference or reallocations from wheat to rice could be associated with lower odds ratios of overall and abdominal fat accumulation, especially for individuals with normal BMI. However, further studies are needed to elucidate the regional specificity of these associations.

## Figures and Tables

**Figure 1 nutrients-14-05243-f001:**
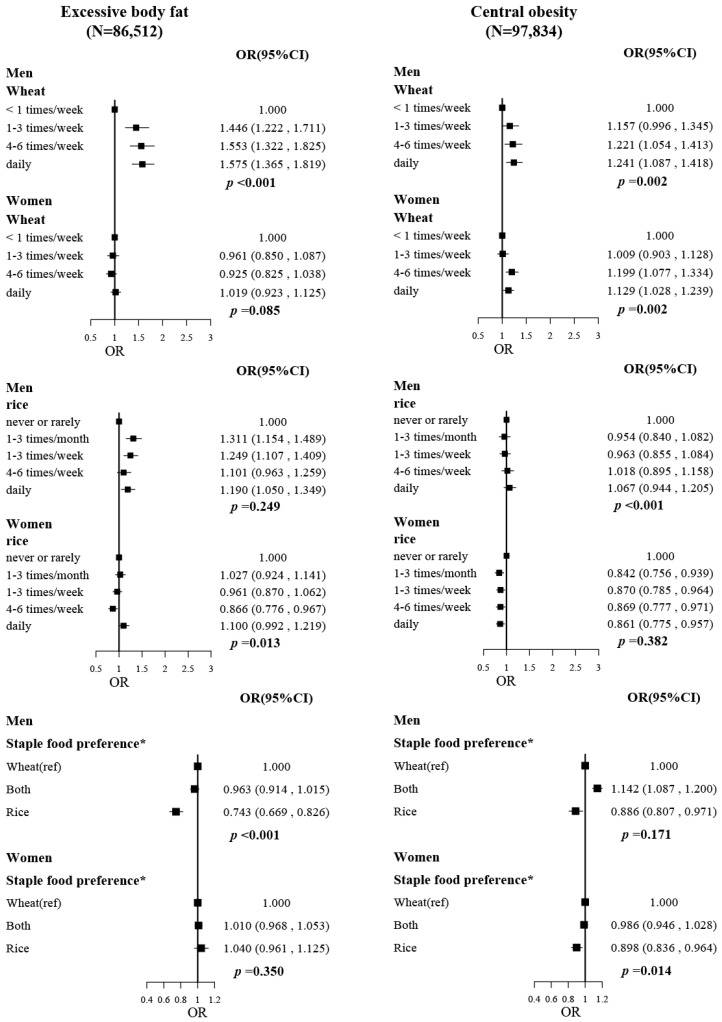
The associations of wheat intake, rice intake, and staple food preference with excessive body fat and central obesity. The “N” on the top of the figure is the total number of participants included in the analysis. Models were fitted based on model 3, with adjustment of age, province, ethnicity, drinking status, smoking status, family income, education level, and physical activity (MET-hr/day). * Staple food preference: rice, consuming rice every day or 4–6 times per week, and consuming wheat less than 4–6 times per week; wheat, consuming wheat every day or 4–6 times per week, and consuming rice less than 4–6 times per week; both, the rest of the participants, which means they consumed rice and wheat with similar frequency.

**Figure 2 nutrients-14-05243-f002:**
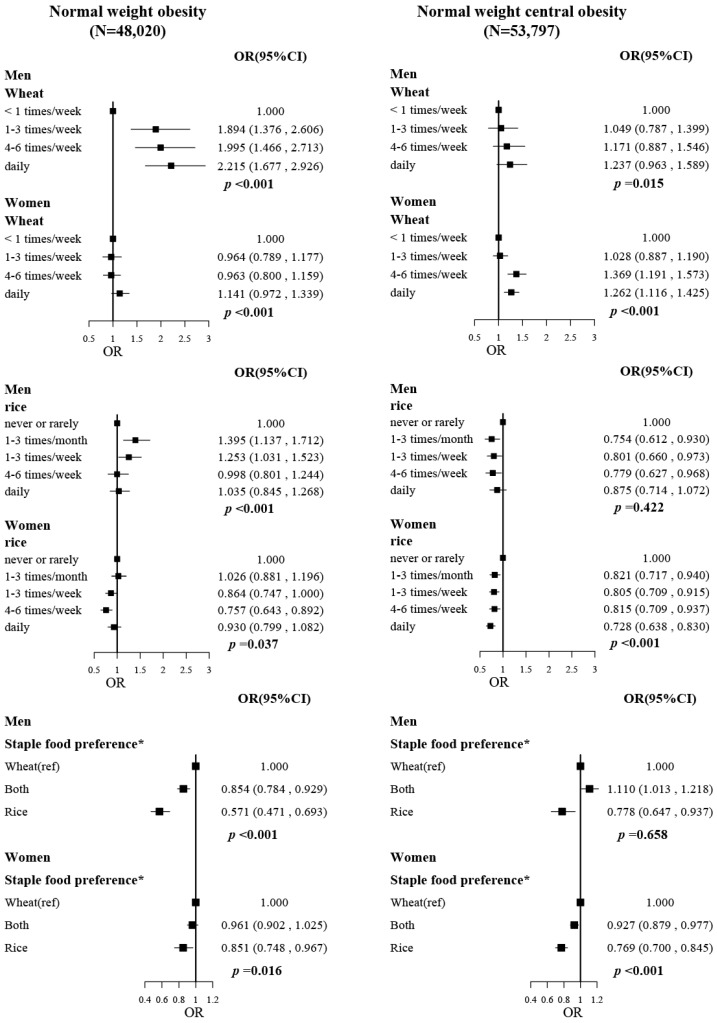
The associations of wheat intake, rice intake, and staple food preference with normal-weight obesity and normal-weight central obesity. The “N” on the top of the figure is the total number of participants included in the analysis. Models were fitted based on model 3, with adjustment of age, province, ethnicity, drinking status, smoking status, family income, education level, and physical activity (MET-hr/day). * Staple food preference: rice, consuming rice every day or 4–6 times per week and consuming wheat less than 4–6 times per week; wheat, consuming wheat every day or 4–6 times per week and consuming rice less than 4–6 times per week; both, the rest of the participants, which means they consumed rice and wheat with similar frequency.

**Table 1 nutrients-14-05243-t001:** Characteristics of the participants by staple food preference and sex.

	Men				Women			
	Wheat	Both	Rice	*p*	Wheat	Both	Rice	*p*
N	21,145	18,111	3340		32,273	25,737	5234	
Age (years, mean (SD))	53.90 (10.81)	51.97 (13.02)	49.02 (15.31)	<0.001	53.12 (9.71)	52.07 (11.94)	48.93 (14.77)	<0.001
Ethnicity								
Han	13,457 (63.64)	15,073 (83.23)	3202 (95.87)	<0.001	21,432 (66.41)	21,008 (81.63)	4994 (95.41)	<0.001
Hui	1768 (8.36)	1766 (9.75)	54 (1.62)		2045 (6.34)	2798 (10.87)	114 (2.18)	
Uygur	4599 (21.75)	864 (4.77)	28 (0.84)		7337 (22.73)	1352 (5.25)	62 (1.18)	
Others	1321 (6.25)	408 (2.25)	56 (1.68)		1459 (4.52)	579 (2.25)	64 (1.22)	
Education								
Primary school or less	10,097 (47.75)	5547 (30.63)	866 (25.93)	<0.001	21,081 (65.32)	11,493 (44.66)	2103 (40.18)	<0.001
Middle or high school	9333 (44.14)	7290 (40.25)	965 (28.89)		10,297 (31.91)	9353 (36.34)	1269 (24.25)	
College or university	1715 (8.11)	5274 (29.12)	1509 (45.18)		895 (2.77)	4891 (19.00)	1862 (35.58)	
Regular alcohol drinking ^a^	2773 (13.11)	3615 (19.96)	976 (29.22)	<0.001	165 (0.51)	345 (1.34)	251 (4.80)	<0.001
Regular smokers ^b^	8677 (41.04)	6570 (36.28)	1257 (37.63)	<0.001	144 (0.45)	144 (0.56)	53 (1.01)	<0.001
Physical activity ^c^								
Low	7487 (35.41)	7547 (41.67)	1041 (31.17)	<0.001	13,451 (41.68)	11,812 (45.90)	1821 (34.79)	<0.001
Middle	5173 (24.46)	5467 (30.19)	1262 (37.78)		9074 (28.12)	8020 (31.16)	2005 (38.31)	
High	8485 (40.13)	5097 (28.14)	1037 (31.05)		9748 (30.20)	5905 (22.94)	1408 (26.90)	
Family income								
<20,000 yuan/year	8765 (41.45)	5309 (29.31)	814 (24.37)	<0.001	14,285 (44.26)	8156 (31.69)	1295 (24.74)	<0.001
20,000–50,000 yuan/year	7869 (37.21)	5451 (30.10)	696 (20.84)		12,542 (38.86)	8642 (33.58)	1326 (25.33)	
≥50,000 yuan/year	3985 (18.85)	6285 (34.70)	1471 (44.04)		4684 (14.51)	7131 (27.71)	1939 (37.05)	
Do not know	526 (2.49)	1066 (5.89)	359 (10.75)		762 (2.36)	1808 (7.02)	674 (12.88)	
Province								
Shaanxi	6626 (31.34)	6927 (38.25)	2597 (77.75)	<0.001	12,263 (38.00)	9238 (35.89)	3932 (75.12)	<0.001
Gansu	5062 (23.94)	3242 (17.90)	232 (6.95)		6765 (20.96)	4226 (16.42)	252 (4.81)	
Qinghai	616 (2.91)	281 (1.55)	65(1.95)		1025 (3.18)	547 (2.13)	142 (2.71)	
Ningxia	1074 (5.08)	4552 (25.13)	28 (0.84)		1471 (4.56)	6907 (26.84)	73 (1.39)	
Xinjiang	7767 (36.73)	3109 (17.17)	418 (12.51)		10,749 (33.31)	4819 (18.72)	835 (15.95)	
Excessive body fat	8833 (47.13)	6545 (51.05)	648 (34.27)	<0.001	15,181 (50.67)	9598 (48.21)	1347 (42.24)	<0.001
BMI > 25 kg/m^2^	9102 (43.12)	8461 (46.79)	1369 (41.12)	<0.001	13,064 (40.55)	9777 (38.03)	1666 (31.89)	<0.001
Central obesity	8147 (40.28)	6833 (43.54)	887 (34.18)	<0.001	20,796 (65.65)	14,732 (62.95)	2285 (53.84)	<0.001

Values are numbers (percentages) unless stated otherwise. ^a^ Regular alcohol drinking was defined as drinking every month or every week. ^b^ Regular smokers were defined as smoking on most days or every day. ^c^ Physical activity was divided into three groups according to the metabolic equivalent of task hours (MET-hr)/day: low, <13.33; middle, 13.33–26.55; high, 26.55 or higher [30]. The *p* values were calculated by χ^2^ test for categorical variables or by ANOVA for continuous variables.

**Table 2 nutrients-14-05243-t002:** The associations of replacing wheat intake (5 times/week) for equivalent frequency of rice with excessive body fat, central obesity, normal-weight obesity, and normal-weight central obesity by sex.

	Men		Women	
	OR (95%CI)	*p*	OR (95%CI)	*p*
Excessive body fat				
Model 1	0.852 (0.787, 0.924)	<0.001	0.958 (0.901, 1.018)	0.166
Model 2	0.807 (0.743, 0.877)	<0.001	0.995 (0.935, 1.060)	0.883
Model 3	0.806 (0.743, 0.876)	<0.001	0.999 (0.939, 1.064)	0.981
Central obesity				
Model 1	1.118 (1.039, 1.203)	0.003	0.840 (0.793, 0.890)	<0.001
Model 2	0.982 (0.910, 1.059)	0.634	0.916 (0.864, 0.973)	0.004
Model 3	0.975 (0.904, 1.052)	0.518	0.918 (0.865, 0.975)	0.005
Participants with normal weight				
Normal weight obesity				
Model 1	0.640 (0.556, 0.737)	<0.001	0.844 (0.765, 0.931)	<0.001
Model 2	0.635 (0.550, 0.733)	<0.001	0.837 (0.757, 0.926)	<0.001
Model 3	0.635 (0.550, 0.733)	<0.001	0.843 (0.762, 0.932)	<0.001
Normal weight central obesity				
Model 1	1.028 (0.895, 1.181)	0.694	0.753 (0.698, 0.812)	<0.001
Model 2	0.907 (0.785, 1.047)	0.183	0.793 (0.734, 0.857)	<0.001
Model 3	0.902 (0.780, 1.043)	0.163	0.795 (0.736, 0.859)	<0.001

Model 1 adjusted for age, province, and ethnicity; Model 2 additionally adjusted for drinking status, smoking status, family income, and education level; Model 3 further adjusted for physical activity (MET-hr/day).

## Data Availability

Data described in the article are available on reasonable request from the corresponding author.

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
