# Peer review of "Staple Food Preference and Obesity Phenotypes: The Regional Ethnic Cohort Study in Northwest China"

_nutrients, 2022, doi:10.3390/nu14245243_

Round 1

Reviewer 1 Report

As the authors note, "determinants of specific adiposity phenotypes remain unclear." The authors aim to evaluate dietary preference patterns with regards to wheat and rice--higher wheat in the north of China and rice in the south. Overwt/obesity is a significant public health issue and may be related to this dietary preference so this is evaluated. This nutritional information has significant public health relevance--the information is an early step along the way to improving health in this population.

Strengths:

--very large study

--statistical methods are sound, robust, appropriate in general

--both the aim of the study and the results are of interest

Weaknesses/questions/comments to address:

--Author s state that the food frequency questionnaire if verified--what questionnaire was used and how was if validated? Authors to please add this clarifying detail to enhance the strength of the paper

--food consumption frequency is equated to wheat vs. rice preference; it may not be accurate to assume that this pattern of consumption is only due to food peference/choice. If there is further information what can be provided to support this link this would be hepful information towards taking the follow up next steps based on the findings of this paper. If the investigators might be interested in changing dietary intake patterns to improve health, is this observation really about preference or is it about access?

--as the investigative team notes, reliance on FFQ is problematic (subjective, reliance on memory, open to bias)

--study is cross-sectional as the authors note

--Some would suggest defining overwt in this population at a BMI>= 23; why did the team choose the level of 25 instead of a lower threshold as put forth by entities like WHO and the NIH? Please add further information/discussion regarding this choice.

--for Table 1 data, regarding the ethanol categories, were there only 2 (at least weekly vs. monthly and nothing in between, and no 0 consumption category?) Please clarify/detail the ethanol categories further.

--Similarly to the ethanol question, for the tobacco question, what does "usually" mean? Is usually the same as everyday or is it meant as a separate choice?  How many tob categories were there? Please clarify/detail the tob categories further. 

--this reviewer very much appreciated the authors' discussion, including the comments that their adiposity-grain preference associational observations may not be about the grains themselves but other foods that tend to be eaten with them. These results will spark interesting follow up studies.

Author Response

Dear reviewers and Editor,

Thanks to all the reviewers and the editor for the constructive and helpful comments to our work. We have revised our manuscript extensively to clarify important findings and conclusions we made. We have addressed the comments provided by the reviewers, point-by-point. All changes are marked in yellow in the revised manuscript.

Reviewer 1

As the authors note, "determinants of specific adiposity phenotypes remain unclear." The authors aim to evaluate dietary preference patterns with regards to wheat and rice--higher wheat in the north of China and rice in the south. Overwt/obesity is a significant public health issue and may be related to this dietary preference so this is evaluated. This nutritional information has significant public health relevance--the information is an early step along the way to improving health in this population.

Strengths:

--very large study

--statistical methods are sound, robust, appropriate in general

--both the aim of the study and the results are of interest

We apricate reviewer’s comments!

Weaknesses/questions/comments to address:

--Authors state that the food frequency questionnaire if verified--what questionnaire was used and how was it validated? Authors to please add this clarifying detail to enhance the strength of the paper.

RESPONSE: We agree with the reviewer’s suggestion. It has been revised as follows:

Lines 90-91: “We used an adapted version of the food frequency questionnaire in the China Kadoorie Biobank, which was showed to have good validity and reproducibility (Nutrients 2022, 14, doi:10.3390/nu14040794).”

--food consumption frequency is equated to wheat vs. rice preference; it may not be accurate to assume that this pattern of consumption is only due to food peference/choice. If there is further information what can be provided to support this link this would be helpful information towards taking the follow up next steps based on the findings of this paper. If the investigators might be interested in changing dietary intake patterns to improve health, is this observation really about preference or is it about access?

RESPONSE: Thanks for the comments! Both rice and wheat powder (products) is easily available in China, with similar affordable prices. Therefore, we believe the consumption pattern is largely due to personal habit or preference.

--as the investigative team notes, reliance on FFQ is problematic (subjective, reliance on memory, open to bias).

--study is cross-sectional as the authors note.

RESPONSE: We agree the comments and we have stated this in the limitation, as follows:

Lines 287-288: “Several limitations should be considered in the present study. Firstly, while the data of dietary intake was selfreported and recall bias could be an issue.”

lines 290-291: “Secondly, our study can’t ensure the causality of the observed associations due to the cross-sectional design.”

--Some would suggest defining overwt in this population at a BMI>= 23; why did the team choose the level of 25 instead of a lower threshold as put forth by entities like WHO and the NIH? Please add further information/discussion regarding this choice.

RESPONSE: Thanks for the comments! We defined BMI within the normal range (18.5-24.9 kg/m2) according to the definition of WHO, which could make our study comparable with other studies.

--For Table 1 data, regarding the ethanol categories, were there only 2 (at least weekly vs. monthly and nothing in between, and no 0 consumption category?) Please clarify/detail the ethanol categories further.

--Similarly to the ethanol question, for the tobacco question, what does "usually" mean? Is usually the same as every day or is it meant as a separate choice? How many tob categories were there? Please clarify/detail the tob categories further.

RESPONSE: Thanks for the comments! There are four categories for alcohol drinking: never or rarely, occasionally, monthly, and weekly; and four smoking categories: never, occasionally, usually, and every day. To simplify the presentation of results, like many studies, these four categories were reclassified into two. Smoking usually means smoking in most of days but not every day in the questionnaire. We have revised it in the notes of Table 1, as “Regular smokers were defined as smoking in most of days or every day”, and in lines 115-117, as “with further adjustment of alcohol drinking (every week or every month, occasionally or never), smoking status (every day or in most of days , occasionally or never)” Besides, we found the associations were not changed substantially, when alcohol drinking with four categories and smoking status with four categories were used in the models. As it shown in the figure below: (author-coverletter-24398074.v1.docx)

--this reviewer very much appreciated the authors' discussion, including the comments that their adiposity-grain preference associational observations may not be about the grains themselves but other foods that tend to be eaten with them. These results will spark interesting follow up studies.

Thanks for reviewer’s comments!

Reviewer 2 Report

The authors present a study on the relationship between staple food preference (wheat, rice or combination) and obesity measures. They conclude that those with wheat as a staple food preference have higher risk of obesity. The study is thorough and is based on a large data set and I therefore think it is appropriate for publication in nutrients, but there are some things which needs to be fixed before the paper can be accepted for publication

In section 2.4 statistical analysis reference 28 is used for the substitution method. I do not agree that this is a good reference for this method, it is merely an application of the method, and not an explanation. Please find a reference with more explanation of this tool.

Figure 1 and 2: What are the numbers in N= for each obesity phenotype modelled?

For each part of the figure, an N is given. Is this the number of people with the given phenotype or is this the number of observations included in the analysis? The total data size as I read it is 105 840 participants (page 2). For the model on Central obesity (Figure 1), N = 97 834 which mean that more than 90% have central obesity and the logistic regression model can potentially be biased (close to a rare event case). In section 2.4 three models are described. Please indicate in the figure legend if the results are based on model 1, 2 or 3.

Supplementary table S1 and S2 and description in text:

The number and percentages do not match. In S1, for the Ningxia province, it says N=22 and 78.57% this cannot be correct.

The statistical analysis does not describe the models applied here, as the OR for wheat is 1.0 for all models and provinces, I assume the model is formulated differently than the models presented in Figure 1 and 2. Moreover, the text describing these results is unclear:

Line 201-202: Similar inverse associations of staple food preference (…) with extensive body fat in men were observed in 4 of 5 provinces… (Table S1)”

In table S1, the OR confidence intervals are below one only for the two provinces Shaanxi (only rice),  and Gansu (both and rice) and above one for Ningxhia (as mentioned in the text). Hence, I do not understand how they conclude with 4 of 5 provinces. For table S2 (woman) a significant association is identified only for the Shaanxi province (correctly described in the paper).

P-values, significance, and effect sizes

Throughout the paper the authors refer to trend when reporting p-values, regardless of the p-values and if they are low (p<0.001) or high (p=0.382). Common practice is to use trend for nearly significant results such as p<0.1. Therefore, I would like that the authors do not use “trend” on the p-values.

However, in this case with such a large data set, I find that reporting p-values is not of great interest, as even small differences can lead to a “significant” p-value. It is OK to have p-values in the tables, but for the discussion and description of results it is in such cases better to focus on the effect size, i.e. the value of the OR, and what this can mean and relate to the number of people relevant for the group. In my opinion the effect sizes here are small. There are no clear-cut rules for small/large effect sizes, but for odds ratio, some references use OR < 1.68 as small effect (or alternative OR > 0.6 for the other way), whereas others use OR<1.44 (or 0.69 for the other way). Very few of the reported confidence intervals are exceeding these values, hence the results are perhaps not so important

Conclusions:

The authors conclude that having wheat as the preferred stable food is associated with higher chance of obesity. Considering uncertainties in conclusions drawn from table S1 and S2, large variations between provinces and small effect sizes I think the authors need to modify and present results accordingly and modify their conclusion to “there could be”, also since they as they point out have not considered the caloric intake.

Final comment: The authors state in the second line of the conclusion that there is “lower risk of overall and abdominal …” In general, the odds-ratio and risk are not identical and cannot be used interchangeably. Consider rewriting of this statement. Reference:  https://www.ncbi.nlm.nih.gov/pmc/articles/PMC4640017/

Author Response

Dear reviewers and Editor,

Thanks to all the reviewers and the editor for the constructive and helpful comments to our work. We have revised our manuscript extensively to clarify important findings and conclusions we made. We have addressed the comments provided by the reviewers, point-by-point. All changes are marked in yellow in the revised manuscript.

Reviewer 2

The authors present a study on the relationship between staple food preference (wheat, rice or combination) and obesity measures. They conclude that those with wheat as a staple food preference have higher risk of obesity. The study is thorough and is based on a large data set and I therefore think it is appropriate for publication in nutrients, but there are some things which needs to be fixed before the paper can be accepted for publication

Thanks for the reviewer’s kind comments!

In section 2.4 statistical analysis reference 28 is used for the substitution method. I do not agree that this is a good reference for this method, it is merely an application of the method, and not an explanation. Please find a reference with more explanation of this tool.

RESPONSE: Thanks for reviewer’s suggestions! We have updated the reference with detailed explanation of the method (Am J Epidemiol 2009, 170, 519-527, doi:10.1093/aje/kwp163).

Figure 1 and 2: What are the numbers in N= for each obesity phenotype modelled?

For each part of the figure, an N is given. Is this the number of people with the given phenotype or is this the number of observations included in the analysis? The total data size as I read it is 105 840 participants (page 2). For the model on Central obesity (Figure 1), N = 97 834 which mean that more than 90% have central obesity and the logistic regression model can potentially be biased (close to a rare event case). In section 2.4 three models are described. Please indicate in the figure legend if the results are based on model 1, 2 or 3.

RESPONSE: Thanks for comments! N is the total number of participants in the analysis. For example, N=97834 in Figure 1 means that 97834 participants were included to assess the associations for central obesity. We have indicated it and model information in the legend of Figure 1 and Figure 2, as follows:

Lines 163-166 and 182-185: “The “N” on the top of figure is the total number of participants included in the analysis. Models were fitted based on model 3, with adjustment of age, province, ethnic, drinking status, smoking status, family income, and education level and physical activity (MET-hr /day).”

Supplementary table S1 and S2 and description in text:

The number and percentages do not match. In S1, for the Ningxia province, it says N=22 and 78.57% this cannot be correct.

RESPONSE: Thanks for comments! N=22 and 78.57% is the total number and proportion of cases among participants with rice preference. In Ningxia, our data showed very small number of people with rice preference.

The statistical analysis does not describe the models applied here, as the OR for wheat is 1.0 for all models and provinces, I assume the model is formulated differently than the models presented in Figure 1 and 2.

RESPONSE: Thanks for comments and we’ve provided more detailed information in statistical analysis part, as follows:

Lines 121-122: “In stratification analyses, categorical variables for stratifying were not adjusted in the models correspondingly”.

Moreover, the text describing these results is unclear: Line 201-202: Similar inverse associations of staple food preference (…) with extensive body fat in men were observed in 4 of 5 provinces… (Table S1)”. In table S1, the OR confidence intervals are below one only for the two provinces Shaanxi (only rice), and Gansu (both and rice) and above one for Ningxia (as mentioned in the text). Hence, I do not understand how they conclude with 4 of 5 provinces. For table S2 (woman) a significant association is identified only for the Shaanxi province (correctly described in the paper).

RESPONSE: Thanks for comments and we are sorry for this misleading. We have revised it as follows:

Lines 206-208: “Inverse associations or trend of staple food preference (from wheat, both wheat and rice, to rice) with excessive body fat in men were observed in Shaanxi and Gansu, while in Ningxia, a positive association was observed, and no significant association in Xinjiang (Table S1)”.

P-values, significance, and effect sizes:

Throughout the paper the authors refer to trend when reporting p-values, regardless of the p-values and if they are low (p<0.001) or high (p=0.382). Common practice is to use trend for nearly significant results such as p<0.1. Therefore, I would like that the authors do not use “trend” on the p-values. However, in this case with such a large data set, I find that reporting p-values is not of great interest, as even small differences can lead to a “significant” p-value. It is OK to have p-values in the tables, but for the discussion and description of results it is in such cases better to focus on the effect size, i.e. the value of the OR, and what this can mean and relate to the number of people relevant for the group. In my opinion the effect sizes here are small. There are no clear-cut rules for small/large effect sizes, but for odds ratio, some references use OR < 1.68 as small effect (or alternative OR > 0.6 for the other way), whereas others use OR<1.44 (or 0.69 for the other way). Very few of the reported confidence intervals are exceeding these values, hence the results are perhaps not so important.

RESPONSE: Thanks for comments and we agree with the reviewer’s suggestions. Descriptions emphasis on p-value have been modified correspondingly.

In Figure 1 and Figure 2, we have removed the “Trend”.

In the results part, we have removed descriptions of trend p in the lines 147-149, 155-157, and 1731-174; and descriptions of p-values in the lines 190-191.

In discussion part, lines 257-259, the sentence “inverse trend of staple food preference from wheat, both wheat and rice, to rice preference was merely significant for overall fat accumulation risk in men, but for abdominal fat accumulation risk in women” has been revised as “the protective associations of rice vs. wheat preference were more remarkable for overall fat accumulation in men, and for abdominal fat accumulation in women.”.

Conclusions:

The authors conclude that having wheat as the preferred stable food is associated with higher chance of obesity. Considering uncertainties in conclusions drawn from table S1 and S2, large variations between provinces and small effect sizes I think the authors need to modify and present results accordingly and modify their conclusion to “there could be”, also since they as they point out have not considered the caloric intake.

RESPONSE: Thanks for suggestions! We have revised it in abstract and conclusion part as follows:

Lines 30-32: “Our data suggests that, compared with wheat, rice preference could be associated with lower odds ratios of certain obesity phenotypes in the Northwest Chinese population.”

Lines 298-300: “Our study suggested that, compared with wheat preference, rice preference or reallocations from wheat to rice could be associated with lower odds ratios of overall and abdominal fat accumulation, especially for individuals with normal BMI. However, further studies are need to elucidate the regional specificity of these associations.”

Final comment: The authors state in the second line of the conclusion that there is “lower risk of overall and abdominal …” In general, the odds-ratio and risk are not identical and cannot be used interchangeably. Consider rewriting of this statement. Reference:https://www.ncbi.nlm.nih.gov/pmc/articles/PMC4640017/

RESPONSE: Thanks for suggestions! We have read this paper carefully and have revised the statements in the corresponding parts.

Lines 31 and 299, the “risks” has been revised as “odds ratios”.
